# A Distributed Algorithm for Real-Time Multi-Drone Collision-Free Trajectory Replanning

**DOI:** 10.3390/s22051855

**Published:** 2022-02-26

**Authors:** Bahareh Sabetghadam, Rita Cunha, António Pascoal

**Affiliations:** Laboratory of Robotics and Engineering Systems (LARSyS), ISR/IST, University of Lisbon, 1649-004 Lisbon, Portugal; rita@isr.ist.utl.pt (R.C.); antonio@isr.ist.utl.pt (A.P.)

**Keywords:** distributed trajectory generation, Voronoi diagram, multi-drone applications, real-time replanning

## Abstract

In this paper, we present a distributed algorithm to generate collision-free trajectories for a group of quadrotors flying through a common workspace. In the setup adopted, each vehicle replans its trajectory, in a receding horizon manner, by solving a small-scale optimization problem that only involves its own individual variables. We adopt the Voronoi partitioning of space to derive local constraints that guarantee collision avoidance with all neighbors for a certain time horizon. The obtained set of collision avoidance constraints explicitly takes into account the vehicle’s orientation to avoid infeasiblity issues caused by ignoring the quadrotor’s rotational motion. Moreover, the resulting constraints can be expressed as Bézier curves, and thus can be evaluated efficiently, without discretization, to ensure that collision avoidance requirements are satisfied at any time instant, even for an extended planning horizon. The proposed approach is validated through extensive simulations with up to 100 drones. The results show that the proposed method has a higher success rate at finding collision-free trajectories for large groups of drones compared to other Voronoi diagram-based methods.

## 1. Introduction

Trajectory generation is a key element for the execution of complex autonomous vehicle missions. It can be defined as the computational problem of finding a valid trajectory that guides a vehicle from an initial state to a given final state in an environment with static and/or moving obstacles. In most applications, the main concern, rather than just finding a feasible trajectory between the initial and final states, is to obtain the optimal trajectory with respect to a certain objective function. In such a setting, trajectory generation is formulated as an optimization problem with a cost function that quantifies the accomplishment of mission goals and objectives, and different types of constraints to ensure safety and feasibility of resulting trajectories.

With rapid advances in communication and computational technology, autonomous vehicles continue to take part in more complex missions, and even engage in teams of collaborating vehicles to take on increasingly demanding tasks. This necessitates incorporating intervehicle collision avoidance constraints in the optimization problem to guarantee that generated trajectories for a group of vehicles sharing a common workspace are collision-free. Therefore, for a large group of vehicles, the optimization problem would involve a large number of constraints and decision variables, and the computational cost of solving it centrally can be prohibitively high. To reduce the computational complexity, a multitude of distributed schemes, reviewed below, have been proposed for decomposing the optimization problem into smaller sub-problems that can be solved locally by each vehicle. The major challenge is to ensure that local decisions do also satisfy the coupling collision avoidance constraints. This is mainly addressed by exchanging information among the vehicles on their current states, future input sequences, etc. Depending on the communication strategy, the sub-problems might be solved sequentially or concurrently, with possibly several iterations of optimization and communication to achieve the required performance.

In [1], the collision avoidance constraint, usually expressed in terms of the two-norm of a relative position vector, is approximated by a set of linear constraints. The sub-problem for each vehicle is then formulated as a mixed-integer linear programming (MILP) that includes the vehicle’s individual variables as well as variables of a subset of neighbors. This enables cooperation among vehicles by allowing a vehicle to make feasible perturbations to neighboring vehicles’ decisions. The sub-problems are solved sequentially by each vehicle, and the algorithm iterates over the group of vehicles until convergence, during each cycle of a model predictive control (MPC) scheme.

Sequential convex programming (SCP)-based methods have also been used for solving distributed multiple vehicle trajectory generation problems [2,3]. The work in [4] addresses the infeasiblity of intermediate problems in decoupled-SCP methods, arising from convex approximation of collision avoidance constraints, i.e., linearizing them, and proposes incremental SCP (iSCP), which tightens collision constraints incrementally. Compared to sequential approaches in [5,6,7], that cast the trajectory of anterior vehicles as dynamic obstacles for a posterior vehicle, the methods proposed in [1,4] result in less constrained intermediate problem and faster convergence rate, yet, similar to most MPC-SCP-based methods, they would require the vehicles to exchange a full representation of their decisions to neighboring vehicles over a communication network.

The synchronous approach in [8] extends the distributed MPC (dMPC) scheme in [9] for formation control, based on alternating direction method of multipliers (ADMM), to problems with intervehicle collision avoidance constraints. These constraints are decoupled using separating hyperplanes, which enforces each vehicle to stay within one half-space of a time-varying plane over a certain time horizon. The resulting sub-problems are solved simultaneously by vehicles, while the normal vector and offset shared between a vehicle and a neighbor, for characterizing their separating hyperplane, are updated at each cycle of the dMPC, using the interchanged information about generated trajectories at the previous cycle.

In the decentralized trajectory planner proposed in [10], vehicles replan their trajectories asynchronously, independent of the planning status of other vehicles. At each iteration, a vehicle considers trajectories assigned to neighboring vehicles as constraints, and solves an optimization problem including as decision variables the normal and offset of planes that separate the outer polyhedral representation of its trajectory and those of its neighbors. A check–recheck scheme is then performed to ensure that the generated trajectory does not collide with trajectories other vehicles have committed to during the optimization time. Therefore, to guarantee deconfliction between vehicles, the planner requires a vehicle to broadcast its computed trajectory to its neighboring vehicles at the end of each replanning iteration.

The on-demand approach to local collision avoidance, proposed in [11], imposes constraints only at specific time instances when collisions between a vehicle and its neighbors are predicted. Predicting collisions along a time horizon, however, relies on an accurate knowledge of the neighbors’ future actions which must be communicated at every sampling time. The dMPC scheme in [12] for distributed trajectory generation is based on this predict–avoid paradigm and an event-triggered replanning strategy, and has been shown to result in less conservative trajectories, but at the cost of voiding the collision avoidance guarantees for all time instances over the horizon. To capture the downwash effect of quadrotor’s propellers, the collision avoidance constraint in [12] is modified with a diagonal scaling matrix, which approximates the quadrotor body with a translating ellipsoid elongated along the vertical axis, yet it ignores the quadrotor’s rotational motion.

Reciprocal velocity obstacle (RVO) and its variants have been widely used in distributed collision avoidance [13,14,15,16,17]. At each time step, RVO [13] builds the set of all relative velocities, leading to a collision between a vehicle and its neighbors, and chooses a new constant velocity outside this set, and closest to the desired value, to avoid collisions. Therefore, RVO requires the position and velocity information to be communicated, or sensed, between nearby neighbors. Other variants, such as acceleration velocity obstacle (AVO), which addresses the instantaneous change of velocity in RVO by taking into account acceleration constraints, need further information such as acceleration to be interchanged. Reciprocally-rotating velocity obstacle (RRVO) [18] uses rotation information to mitigate deadlocks caused by symmetries of representing vehicles with translating discs in RVO. It relies on the assumption that neighbors may rotate equally (or equally opposite), bounded by a maximum value, to compute an approximation of swept areas for rotating polygon-shaped vehicles, and uses them for constructing velocity obstacles. A new velocity and rotation is then selected at each time step to avoid collisions.

Another approach to distributed collision avoidance is to construct the Voronoi diagram of the group of vehicles and generate the trajectory for each vehicle so that it is entirely within the vehicle’s Voronoi cell [19,20,21,22]. Since Voronoi cells do not overlap, it can be guaranteed that the generated trajectories are collision-free. To consider the physical size of a vehicle, the modified Voronoi cell used in [23,24] retracts boundary hyperplanes of the general Voronoi cell by a safety radius for disc-shaped vehicles. At each sampling time, upon receiving the relative position information, trajectories are replanned to conform to the updated Voronoi diagram. The resulting sub-problems can be solved simultaneously, in a receding horizon manner, until the vehicles reach their final positions. The Voronoi-based approaches only require the vehicles to know relative positions to neighboring vehicles, and therefore are well suited to applications where vehicles only have relative position sensing and no communication network [25].

In this paper we develop a distributed trajectory generation framework, with low computation and communication demands, for multiple quadrotors flying in (relatively) close proximity to each other. We specifically address the shortcomings of approximating the drone body with a disc (or sphere) for generating feasible collision-free trajectories for large groups of drones. A sphere model, used in most existing distributed collision avoidance schemes, may be overly conservative in confined spaces since it invalidates trajectories whose feasibility depends on the consideration of the flight attitude. Instead, we model the drone body with an ellipsoid, and employ the Voronoi partitioning of space to derive local collision avoidance constraints that take into account the drone’s real size and orientation. The same approach can be integrated into other distributed schemes that utilize separating hyperplanes for decoupling collision avoidance constraints. Yet the main reason for adopting Voronoi diagram is that using time-invariant boundary hyperplanes determined prior to solving a sub-problem, despite being more conservative, can significantly reduce communication and computational load, allowing for higher replanning rates. Incorporating the resulting set of constraints into sub-problems, solved by each vehicle, allows finding collision-free trajectories for guiding a group of drones through confined spaces by proper adjustment of attitude angles. In addition, the obtained set of constraints can be expressed as Bézier curves, and hence can be efficiently evaluated to guarantee that intervehicle collision avoidance requirements are met at any instant of time even over a long planning horizon.

In the proposed synchronous distributed scheme, each vehicle uses the position information of its neighbors, updated at each sampling time, and solves a sub-problem to generate its trajectory inside (a subset of) its Voronoi cell towards the closest point (in the cell) to its goal position. We present an efficient method to compute this point, which is needed to appropriately define the terminal constraint and cost in the sub-problem. A sequence of sub-problems are then solved in a receding-horizon manner until the vehicles reach their goal positions. The simulation results show that the proposed method has a higher success rate at finding collision-free trajectories for larger groups of quadrotors compared to other Voronoi diagram-based methods. In addition, it can effectively reduce the total flight time required to perform point-to-point maneuvers. Furthermore, the computation time of generating those trajectories satisfies timing constraints imposed by real-time applications.

The rest of this paper is organized as follows: In Section 2 we formulate the optimization sub-problem solved by each vehicle. In Section 2.1 we study the differentially flat system describing the drone equations of motion, and parameterize trajectories with Bézier curves. We derive the set of local collision avoidance constraints in Section 2.2, and present an efficient algorithm for finding the closest point in a Voronoi polytope to a goal position in Section 2.3. In Section 3 we obtain the continuity conditions between two adjacent Bézier curve segments, and present a method for evaluating inequalities in Bézier form without discretization. Finally, we provide simulation results in Section 4.

## 2. Problem Formulation

The multiple vehicle trajectory generation problem addressed in this paper can be defined as finding optimal trajectories that act as references to guide a group of vehicles from their initial positions to some desired final positions. The generated trajectories should jointly minimize a cost function, corresponding to the accomplishment of mission goals and objectives, and satisfy a set of local and coupling constraints, so that they are dynamically-feasible and collision-free. For $N_{v}$ vehicles, this problem can be formulated as the following optimal control problem.
(1)$$\begin{array}{ccc} \min_{\begin{array}{c} u_{i}\left(.\right)\\ i\in [N_{v}] \end{array}} & \sum_{i\in [N_{v}]}J\left(x_{i}\left(.\right),u_{i}\left(.\right)\right) & \\ s.t.\; & {\dot{x}}_{i}\left(t\right)=f\left(x_{i}\left(t\right),u_{i}\left(t\right)\right) & \left(\mathrm{Dynamics}\right)\\ \; & x_{i}\left(0\right)=x_{i,0} & \left(\mathrm{Initial}\;\mathrm{state}\right)\\ \; & x_{i}\left(t_{f}\right)=x_{i,f} & \left(\mathrm{Final}\;\mathrm{state}\right)\\ \; & c\left(x_{i}\left(t\right),x_{j}\left(t\right)\right)\leq 0\;j\in \left[N_{v}\right]\backslash \left\{i\right\} & \left(\mathrm{collision}\;\mathrm{avoidance}\right)\\ \; & x_{i}\left(t\right)\in X_{i} & \left(\mathrm{State}\;\mathrm{Constraints}\right)\\ \; & u_{i}\left(t\right)\in U_{i} & \left(\mathrm{Input}\;\mathrm{constraints}\right) \end{array}$$
where $\left[N_{v}\right]=\left\{1,\cdots ,N_{v}\right\}$. The cost to be minimized is the sum of the vehicles’ individual costs, *J*, given by the functional,
(2)$$J\left[u_{i}\left(.\right)\right]=\int_{0}^{t_{f}}L\left(x_{i},u_{i}\right)dt$$
where $x_{i}\left(t\right)\in R^{n_{x}}$ and $u_{i}\left(t\right)\in R^{n_{u}}$ are the state and the input vectors of the vehicle’s model described by an ODE, and $x_{i,0}$ and $x_{i,f}$ are the initial and final values of the state of the *i*-th vehicle, respectively. $X_{i}$ and $U_{i}$ denote the set of admissible states and inputs for the *i*-th vehicle derived from limits imposed by vehicle dynamics and the surrounding environment.

In order to reduce the computational complexity of solving (Equation 1) for larger $N_{v}$ with increased numbers of constraints and variables, one can divide the problem into a set of small-scale sub-problems. Here, the sub-problems are formulated such that each involves only a vehicle’s individual costs and constraints, and hence can be solved independently by the vehicle. The sub-problems must include constraints to ensure that the trajectory generated locally by a vehicle does satisfy the coupling collision avoidance constraints.

The key idea to ensure intervehicle collision avoidance is to decompose the space into non-overlapping regions, provided by a Voronoi diagram, and generate the trajectory for each vehicle such that it is entirely within its partition. The Voronoi diagram is updated at each sampling time according to the relative positions of vehicles, and a sequence of sub-problems is solved in a receding horizon manner until the vehicles reach their final positions. For the *i*-th vehicle, the problem that has to be solved at the time instant $t_{k}$ can be formulated as
(3)$$\begin{array}{ccc} \min_{\begin{array}{c} x_{i,k}\left(.\right),u_{i,k}\left(.\right) \end{array}} & J(x_{i,k}\left(.\right),u_{i,k}\left(.\right)) & \\ s.t.\; & {\dot{x}}_{i,k}\left(t\right)=f\left(x_{i,k}\left(t\right),u_{i,k}\left(t\right)\right) & \left(\mathrm{Dynamics}\right)\\ \; & x_{i,k}\left(t_{k}\right)={\hat{x}}_{i,k} & \left(\mathrm{Initial}\;\mathrm{state}\right)\\ \; & x_{i,k}\left(t\right)\in C_{i,k}\left({\bar{x}}_{k}\right) & \left(\mathrm{collision}\;\mathrm{avoidance}\right)\\ \; & x_{i,k}\left(t\right)\in X_{i,k} & \left(\mathrm{State}\;\mathrm{Constraints}\right)\\ \; & u_{i,k}\left(t\right)\in U_{i,k} & \left(\mathrm{Input}\;\mathrm{constraints}\right) \end{array}$$
where $x_{i,k}\left(t\right)$ and $u_{i,k}\left(t\right)$ are the state and the input profiles of the vehicle over the time interval $\left[t_{k},t_{k}+t_{h}\right]$, with $t_{h}$ being the planning horizon, and ${\hat{x}}_{i,k}$ denotes its state at the time instant $t_{k}$. The cost function in the above sub-problem is modified as
(4)$$J\left[u_{i,k}\left(.\right)\right]=\int_{t_{k}}^{t_{k}+t_{h}}L\left(x_{i,k},u_{i,k}\right)dt+\phi \left(x_{i,k}\left(t_{k}+t_{h}\right)\right)$$
where the second term is added to penalize the distance, at $t_{k}+t_{h}$, to the point in the Voronoi partition that is closest to the goal position.

In the optimization problem (Equation 3), $C_{i,k}$ may denote the Voronoi partition assigned to the *i*-th vehicle. The Voronoi diagram is updated for each sub-problem according to the vehicles’ configuration at each time instant $t_{k}$, i.e., ${\bar{x}}_{k}={\left\{{\hat{x}}_{j,k}\right\}}_{j\in [N_{v}]}$. Since Voronoi partitions are disjoint and the assigned trajectory to each vehicle for the time horizon $t_{h}$ is contained within its partition, it can be guaranteed that there is no collision between the trajectories over the time interval $\left[t_{k},t_{k}+t_{h}\right]$.

The distributed trajectory generation framework is summarized in Algorithm 1. In Section 2.2, we study the Voronoi diagram for a group of vehicles and modify $C_{i,k}$ to explicitly take into account the orientation while generating collision-free trajectories for multiple drones.
**Algorithm 1 **Distributed Trajectory Generation Framework1:k=02:${\hat{x}}_{i,0}\leftarrow$ Initial position of the *i*-th vehicle3:**repeat**4:    Receive position information from neighbors5:    Broadcast own position to neighbors6:    Update Voronoi partition7:    Compute the closest point in the Voronoi partition to    the goal position      ▹ Section 2.38:    Set the cost function (Equation 4)9:    Set the constraints    ▹ Section 2.2 and Section 3.110:    Solve the optimization sub-problem11:**until**${\hat{x}}_{i,k}=x_{i,f}$.

### 2.1. Quadrotor Model

In this paper, the simplified quadrotor equations of motion are described by
(5)$$m\ddot{p}=mge_{3}+f,$$
where $p\in R^{3}$ is the position and *m* is the mass of the quadrotor. In addition, $g=9.8\;\frac{m}{s^{2}}$ is the gravitational acceleration, and $e_{3}={\left[0\;0\;1\right]}^{T}$. The first term on the right-hand side of (Equation 5) is gravity in the $z_{I}$ direction, and the second term, $f\in R^{3}$, is the thrust force aligned with the body’s z-axis.
(6)$$f=-T^{I}z_{B}$$
where T∈R is the net thrust, ${}^{I}z_{B}=R^{B}z_{B}=Re_{3}$ is the body frame z-axis expressed in {I}, and $R\equiv {}_{B}^{I}R\in SO\left(3\right)$ is the rotation matrix from the body frame {B}, centered at the quadrotor’s center of gravity, to the fixed inertial frame {I}. For simplicity, we drop the superscript ${}^{I}$ and consider $z_{B}={}^{I}z_{B}$. Figure 1 is a graphical representation of the quadrotor and the associated reference frames.

#### Trajectory Parametrization

The quadrotor dynamics (Equation 5) with the four inputs is differentially flat [26], and therefore the state and the input of the system can be expressed as functions of the flat outputs and a finite number of its derivatives. The position vector together with the yaw angle can be selected as flat outputs of the system. Here, $p\in R^{3}$ is parameterized as a Bézier curve, given by
(7)$$\begin{array}{c} p\left(\tau \right)=\sum_{l=0}^{n}{\bar{p}}_{l}B_{l,n}\left(\tau \right), \end{array}$$
where ${\bar{p}}_{l}\in R^{3}$ are the control points, $B_{l,n}$ are Bernstein basis polynomials of degree *n*, and τ∈[0,1] is defined as
(8)$$\tau =\frac{t}{t_{f}}$$

The linear velocity, $v=\dot{p}$, and linear acceleration, $a=\ddot{p}$, can be expressed as parametric Bézier curves through the first and second derivative of p with respect to time, yielding
(9)$$\begin{array}{ccc} & v\left(\tau \right)=\sum_{l=0}^{n-1}{\bar{v}}_{l}B_{l,n-1}\left(\tau \right) & \\ \; & a\left(\tau \right)=\sum_{l=0}^{n-2}{\bar{a}}_{l}B_{l,n-2}\left(\tau \right) \end{array}$$
where the control points ${\bar{v}}_{l}$ and ${\bar{a}}_{l}$ are obtained as
(10)$$\begin{array}{cccc} & {\bar{v}}_{l}=\frac{n}{t_{f}}\left({\bar{p}}_{l+1}-{\bar{p}}_{l}\right) & \;l=0,\cdots ,n-1 & \\ \; & {\bar{a}}_{l}=\frac{n(n-1)}{t_{f}^{2}}\left({\bar{p}}_{l+2}-2{\bar{p}}_{l+1}+{\bar{p}}_{l}\right) & \;l=0,\cdots ,n-2 \end{array}$$

The thrust *T* and rotation matrix *R* can also be expressed as functions of the flat output and its derivatives. The net thrust *T* can be written as
(11)$$T=m∥\ddot{p}-ge_{3}∥.$$

Assuming that the rotation matrix $R=[x_{B},y_{B},z_{B}]$ is parameterized by the *Z*-*Y*-*X* Euler angles $\lambda ={\left[\phi ,\theta ,\psi \right]}^{T}$ as
(12)$$R=R_{z}\left(\psi \right)R_{y}\left(\theta \right)R_{x}\left(\phi \right),$$
then the columns of the rotation matrix are extracted from
(13)$$z_{B}=\frac{ge_{3}-\ddot{p}}{∥ge_{3}-\ddot{p}∥}\;x_{B}=\frac{r\times z_{B}}{∥r\times z_{B}∥},\;y_{B}=z_{B}\times x_{B}$$
where the unit vector r is defined as
(14)$$r={\left[-\sin\psi ,\cos\psi ,0\right]}^{T}$$

The above equations declare that the vehicle’s orientation can be fully determined from the second derivative of the trajectory and the yaw angle. As mentioned before, the yaw angle ψ is a component of the flat output, and therefore it can be controlled independently without affecting the trajectory generation. Using the differential flatness property of the system, trajectories consistent with dynamics can be planned in the space of flat outputs, where (Equation 5) is trivially satisfied and the original input and state constraints are transformed into constraints on the flat output and its derivatives.

### 2.2. Collision Avoidance

In this section, we present a Voronoi diagram-based approach to decoupling intervehicle collision avoidance constraints. Although the presence of obstacles, interpreted as non-decision-making agents, is not explicitly considered here, incorporating vehicle–obstacle collision avoidance constraints into the problem simply amounts to taking into account the obstacles’ position when updating the Voronoi partition (step 6 of Algorithm 1).

The widely used approach in the literature to avoiding collisions with obstacles in the environment is to model the drone body as a sphere with radius $r_{D}$, and then simply building the collision-free space, $C_{free}$, by inflating the obstacles with a factor $r_{D}$. As a result, collision-free trajectories can be obtained by enforcing the vehicle, which is now treated as a point in the space, to be inside $C_{free}$ [27]. Considering now the collision avoidance between the *i*-th and *j*-th drones, the corresponding constraint can be derived similarly by
(15)$$\begin{array}{c} ∥p_{i}-p_{j}∥\geq 2r_{D} \end{array}$$
where ∥.∥ denotes the Euclidean distance. Ignoring the real shape and orientation of the drone, and approximating its body with a sphere, invalidates trajectories that are feasible upon considering the flight attitude. For this reason, the above approach might be too conservative for trajectory generation in confined spaces and can even fail to find feasible collision-free trajectories when multiple drones are involved.

Approximating the drone body with an ellipsoid, whose principal axes are aligned with the body frame axes, allows considering the drone orientation while inspecting for collisions against other vehicles. For the *i*-th drone, the ellipsoid, $E_{i}$, centered at the drone position $p_{i}$, is given by
(16)$$E_{i}\equiv \{p\in R^{3}|p=p_{i}+R_{i}\Lambda w,∥w∥\leq 1\}$$
where Λ is a 3×3 diagonal matrix of the form
(17)$$\Lambda =\left[\begin{array}{ccc} r_{D} & 0 & 0\\ 0 & r_{D} & 0\\ 0 & 0 & h_{D} \end{array}\right]$$
with $r_{D}$ and $h_{D}$ being the lengths of the principle semi-major and semi-minor axes, respectively. (See Figure 2).

As proposed in [28], collision avoidance constraints for two ellipsoid-shaped drones can be derived using separating hyperplanes. Denoting by $a\in R^{3}$ and b∈R the normal vector and offset of a hyperplane, respectively, the separating hyperplane for $E_{i}$ and $E_{j}$, defined as $H\equiv \{p|a^{T}p-b=0\}$, must satisfy
(18)$$\begin{array}{cc} a^{T}p-b\leq 0 & \;\forall p\in E_{i}\\ a^{T}p-b>0 & \;\forall p\in E_{j} \end{array}$$

Since
(19)$$-∥\Lambda R^{T}a∥\leq a^{T}R\Lambda w\leq ∥\Lambda R^{T}a∥$$

The set of inequalities (Equation 18) holds if, and only if,
(20)$$\begin{array}{ccccc} \;\;\;a^{T}p_{i}-b & \geq & \; & ∥\Lambda R_{i}^{T}a∥ & \;\;\;\\ a^{T}p_{j}-b & \leq & - & ∥\Lambda R_{j}^{T}a∥ \end{array}$$

Satisfying the set of constraints (Equation 20) will guarantee that there is no collision between the two ellipsoids $E_{i}$ and $E_{j}$ associated with the *i*-th and *j*-th drones, respectively.

For multiple vehicle trajectory generation, collision avoidance constraints, either in the form of the inequality constraint (Equation 15), for spheres, or the set of constraints (Equation 20), for ellipsoids, must be incorporated in the optimization problem for each pair of vehicles. As the number of vehicles involved in a mission grows, the resulting increase in the number of constraints would inevitably exacerbate the computational issues of finding collision-free trajectories in a centralized manner.

#### Distributed Collision Avoidance

Here, we propose a distributed approach to collision avoidance which takes into account the shape and orientation of a drone. The presented approach uses Voronoi partitioning of space and generates the trajectory of each vehicle such that it is entirely within (a subset of) the vehicle’s Voronoi cell for a time interval $t_{h}$. Since Voronoi cells are pairwise disjoint, and each vehicle only moves inside its Voronoi cell, intervehicle collision avoidance is guaranteed for all future time before $t_{h}$.

Each Voronoi cell in an n-dimensional space is a convex polytope bounded by a number of (n−1)-dimensional convex polytopes. For a group of vehicles in three-dimensional space, the general Voronoi cell of the *i*-th vehicle is defined as
(21)$$V_{i}=\{p\in R^{3}|{p_{ij}}^{T}\left(p-\frac{1}{2}\left(p_{i}+p_{j}\right)\right)\leq 0,\forall j\in \left[N_{v}\right]\backslash \left\{i\right\}\}$$
where $p_{ij}=p_{j}-p_{i}$, and $p_{i}$ and $p_{j}$ are the position of the *i*-th and *j*-th vehicles at the current time instant. Note that $V_{i}$ is the intersection of half-spaces corresponding to hyperplanes with $a=p_{ij}$ and $b=\frac{1}{2}p_{ij}^{T}\left(p_{i}+p_{j}\right)$. An arbitrary point in $V_{i}$ is closer to the *i*-th vehicle than any other vehicle [22], i.e.,
(22)$$∥p-p_{i}∥\leq ∥p-p_{j}∥,\;\forall p\in V_{i}\;\&\;j\neq i$$

The boundary of the Voronoi cell, $\partial V_{i}$, is the union of multiple faces, each of which include points in the space that are equidistant to the *i*-th vehicle and a neighboring vehicle.

In order to account for the size of a vehicle, the buffered Voronoi cell (BVC) proposed in [24] retracts the boundary of the general Voronoi cell by a safety radius, so that if the vehicle’s center is inside the BVC, its body, approximated by a sphere of radius $r_{D}$, will be entirely within its Voronoi cell. The BVC of the *i*-th vehicle, denoted by ${\bar{V}}_{i}$, is defined as
(23)$$\begin{array}{cc} {\bar{V}}_{i}=\{p\in R^{3}|\;p_{ij}^{T}(p-\frac{1}{2}\left(p_{j}+p_{i}\right)) & +r_{D}∥p_{ij}∥\leq 0,\forall j\in \left[N_{v}\right]\backslash \left\{i\right\}\} \end{array}$$

It can be easily shown that BVC is a subset of the general Voronoi cell, i.e., ${\bar{V}}_{i}\subset V_{i}$. In addition, for any two points $p^{\prime }\in {\bar{V}}_{i}$ and $q^{\prime }\in {\bar{V}}_{j}$, $∥p^{\prime }-q^{\prime }∥\geq 2r_{D}$ holds. Therefore, the vehicles are guaranteed to avoid collisions due to the buffer region of $r_{D}$ along $\partial V_{i}$. Figure 3 shows the Voronoi diagram for 10 drones in a collision-free configuration and the BVC for two adjacent drones.

The BVC is defined based on a symmetrical approximation of the vehicle’s body with a translating disc. In order to reduce the conservatism and avoid infeasibility issues due to ignoring the real shape and orientation of the vehicle, we approximate the drone with an ellipsoid (Equation 16), and bearing in mind that
(24)$$R\Lambda^{2}R^{T}=r_{D}^{2}I+\left(h_{D}^{2}-r_{D}^{2}\right)z_{B}z_{B}^{T},$$
we propose $C_{i}$ in problem (Equation 3) to be defined as
(25)$$\begin{array}{cc} C_{i}=\left\{\left(p,\ddot{p}\right)\in R^{6}\right|p_{ij}^{T}\left(p-\frac{1}{2}\left(p_{j}+p_{i}\right)\right) & +∥\Lambda R^{T}p_{ij}∥\leq 0,\forall j\in \left[N_{v}\right]\backslash \left\{i\right\}\} \end{array}$$

If the trajectory of the *i*-th drone $p_{i}\left(t\right)$ satisfies the above set of local collision avoidance constraints for all $t\in [t_{k},t_{k}+t_{h}]$, then the ellipsoid representing the drone body is within the Voronoi cell for the entire time horizon; that is, the ellipsoid centered at $p_{i}$ and aligned with the columns of $R_{i}$ does not intersect the Voronoi boundary, stated mathematically
(26)$$∥\partial E_{i}-\partial V_{i}∥\geq 0$$

Noting that
(27)$$h_{D}∥p_{ij}∥\leq ∥\Lambda R^{T}p_{ij}∥\leq r_{D}∥p_{ij}∥,$$
it can be induced that
(28)$${\bar{V}}_{i}\left(r_{D}\right)\subset proj_{XYZ}C_{i}\subset {\bar{V}}_{i}\left(h_{D}\right)\subset V_{i}$$
where ${\mathrm{proj}}_{XYZ}C_{i}$ is the projection of $C_{i}$ onto the three-dimensional subspace spanned by $e_{1}$, $e_{2}$, and $e_{3}$.

Therefore, incorporating (Equation 25) into the optimization problem (Equation 3) will ensure that the generated trajectories are collision-free while alleviating infeasibility problems by taking orientations into account. Also, since $z_{B}$ is fully obtained from $\ddot{p}$ (Equation 13), the above set of local collision avoidance constraints can be expressed as constraints imposed on Bézier curves. Later, we present an efficient method for evaluating inequalities in Bézier form.

### 2.3. Finding the Closest Point to the Goal Position

As explained above, at each time instant the Voronoi cell $V_{i}$ is updated according to the relative position of the *i*-th vehicle to other vehicles. The optimization problem (Equation 3) is then solved to generate a trajectory, for a time horizon $t_{h}$, that guides the vehicle towards the point in the cell closest to the goal position. This process is repeated until the vehicle reaches its final position. At each sampling time, the closest point must be found prior to solving the trajectory generation problem. Therefore, having an efficient scheme for finding the closest point is critically important to avoid long computational delays between updating the Voronoi cell and replanning the trajectory.

The point in a convex polytope that is closest to a query point q is either q itself or a point on the boundary of the polytope. A naive way to find the closest point in a convex polytope in a three-dimensional space, represented by P=(F,E,V), where F is the set of faces, E is the set of edges, and V is the set of vertices, is to check the distance between $q\in R^{3}$ to all faces, edges, and vertices for finding the minimum. However, for complex polytopes, the computation time is not negligible.

The geometric approach proposed in [24] for a polygon in a two-dimensional space calculates the barycentric coordinates and an angle from the query point to the two vertices of each edge to find the closest point. Since this approach iterates over all edges, its computational complexity can significantly increase as the number of Voronoi neighbors of a vehicle increases. Here, we make use of the Gilbert–Johnson–Keerthi (GJK) distance algorithm and devise an approach that can efficiently determine whether the query point is inside the polytope, i.e., q∈P, in which case the closest point is q itself. Otherwise, the presented algorithm returns the closest feature of *P* to q, and the closest point can be obtained by projecting q onto it. The proposed approach is not limited to distance queries between a point and a polytope, and can also be used when the final constraint in (Equation 3) is relaxed to a small box or sphere around the goal position, and used in conjunction with a terminal cost term.

The GJK distance algorithm or simply GJK algorithm is an iterative algorithm that relies on a support mapping function to incrementally build simplices that are closer to the query point [29]. The algorithm has been extensively used for collision detection between generic convex shapes [30,31]. The original algorithm, however, can be used to compute the minimum distance, and also the respective pair of (closest) points, between two convex shapes [32].

In order to obtain the minimum distance between two general convex sets *A* and *B*, GJK approximates the closest point in the Minkowski difference of the two sets, C=A−B to the origin, denoted by ν(C), with an iterative search. At each iteration, a simplex in *C* is constructed such that it is closer to the origin O than the simplex in the previous iteration. In $R^{3}$, a simplex can be a point, a line, a triangle, or a tetrahedron with 1, 2, 3, and 4 affinely independent vertices.

GJK relies on the so-called support mappings to construct a new simplex. A support mapping function $s_{C}\left(d\right)$ of the convex set *C* maps the vector d to a point in the set, called the support point, according to
(29)$$d^{T}s_{C}\left(d\right)=\max\left\{d^{T}p;p\in C\right\}$$

At each iteration, a support point $w_{k}=s_{C}\left(-\nu_{k}\right)$ is added as a vertex to the current simplex, indicated with $V_{k}$, where $\nu_{k}$ is the closest point of $V_{k}$ to the origin, i.e., $\nu_{k}=\nu \left(\mathrm{conv}\left(V_{k}\right)\right)$. $V_{k+1}$ is then updated such that it only contains the smallest set of vertices that supports $\nu_{k+1}=\nu \left(\mathrm{conv}\left(V_{k}\cup w_{k}\right)\right)$, and earlier vertices that do not back $\nu_{k+1}$ are discarded [30].

It is proved in [30] that in each iteration the new ν is closer to the origin than the previous one, and thus the sequence of $\left\{\nu_{k}\right\}$ converges to the closest point of *C* to the origin. In addition, it is shown that
(30)$$∥\nu_{k}{-\nu \left(C\right)∥}^{2}\leq {∥\nu_{k}∥}^{2}-\nu_{k}^{T}w_{k}$$
which is used to construct the terminating condition of the GJK algorithm for general convex shapes.

The GJK algorithm, as explained above, depends heavily on the computation of $\nu_{k}$ to test the termination condition and to determine the search direction for finding the support point. In each iteration of the algorithm, $\nu_{k}$ must be computed with the *Johnson Distance Subalgorithm* [29] or more robust alternatives such as the signed volume method in [33]. Here, we exploit unique features of polytopes and propose a faster way to evolve simplices in the GJK algorithm without computing $\nu_{k}$ in each iteration.

For polytopes, GJK arrives at the actual ν(C) in a finite number of iterations [30]. The pseudocode in Algorithm 2 describes the GJK distance algorithm for a polygon *P*. In order to find the support point $w_{k}$, we employ a search direction $d_{k}↑↓\nu_{k}$, which is updated in each iteration of the algorithm with S1D, S2D, or S3D subroutines. To update d and *V*, these subroutines, summarized in Algorithms 3–5, inspect the Voronoi regions of the simplex for the one that contains the origin. Figure 4 shows the Voronoi regions of an m-simplex (m=1,2,3) where the origin possibly lies. Once the Voronoi region containing the origin is found, d is determined as a vector from the associated vertex, edge, or face (of the simplex) pointing towards the origin.

The stop criterion for the conditional loop in Algorithm 1 is also constructed using the search direction, offered as
(31)$$d_{k}^{T}\left(w_{k}-v_{1}\right)\leq 0$$
where $v_{1}\in V_{k}$. Considering that $d_{k}^{T}\left(\nu_{k}-v_{1}\right)=0$, we can conclude that the above criterion, for deciding whether $V_{k}$ represents the closest feature of *P* to the origin, is equivalent to the stop criterion in [30] for determining whether $\nu_{k}$ s the closest point, that is
(32)$$∥\nu_{k}{∥}^{2}-\nu_{k}^{T}w_{k}\leq 0⟷d_{k}^{T}\left(w_{k}-v_{1}\right)\leq 0$$

If the inequality (Equation 31) holds, then the closest feature of *P* to the origin can be determined from $V_{k}$. Figure 5 shows all possible outputs of Algorithm 2, which can be a vertex, an edge, or a face of *P* or a tetrahedron inside it, and ν(P) for each of the cases.
**Algorithm 2 **Compute the closest point of *P* to the origin1:v= “Arbitrary point in vert(P)”2:d=−v3:V={v}4:**repeat**5:    $w=s_{P}\left(d\right)$;6:    **if** $d^{T}\left(w-v_{1}\right)\leq 0$ **then**7:        *V* represents the closest feature of *P* to O8:        **return** ν(V)9:    **end if**10:    V←V∪w;11:    [V,d]← CallSmD(V);^1^12:**until**|V|=4;13:*P* contains O14:**return**ν=O^1^ One of the three subroutines S1D, S2D or S3D is called in accordance with |V|.

A support point of a convex polytope can also be computed efficiently. For a polytope *P*, the support point is a vertex of *P*, i.e., $s_{P}\left(d\right)\in vert\left(P\right)$, and we can take $w=s_{P}\left(d\right)=s_{vert(P)}\left(d\right)$, that is,
(33)$$d^{T}s_{P}\left(d\right)=\max\left\{d^{T}v;v\in vert\left(P\right)\right\}$$

Therefore, for polytopes, the support point can be uniquely determined by simply scanning through the list of vertices for the vertex that is the most extreme in the search direction d. Therefore, the computation time is linear in the number of vertices of *P*. For complex polytopes, the vertices adjacency information and the coherence between consecutive calls to support mapping functions can be exploited to find the support point with almost constant time complexity [30].
**Algorithm 3 **Sub-routine for |V|=21:**function**S1D($\left\{v_{2},v_{1}\right\}$) ^1^2:    **if** $v_{1}^{T}v_{12}\geq 0$ **then**3:        $V\leftarrow \{v_{1}\}$4:        $d\leftarrow -v_{1}$5:    **else**6:        $V\leftarrow \{v_{2},v_{1}\}$7:        $d\leftarrow -v_{12}\times v_{1}\times v_{12}$8:    **end if**9:**end function**^1^ The input is the ordered list of vertices, with $v_{1}$ being the last added element to $V_{k}$.

**Algorithm 4 **Sub-routine for |V|=3
1:**function**S2D($\left\{v_{3},v_{2},v_{1}\right\}$)2:    $n_{v_{3}v_{2}v_{1}}=v_{12}\times v_{13}$3:    **if** $v_{1}^{T}\left(n_{v_{3}v_{2}v_{1}}\times v_{12}\right)\geq 0$ **then**4:        [V,d]← S1D($\left\{v_{2},v_{1}\right\}$)5:    **else**6:        **if** $v_{1}^{T}\left(v_{13}\times n_{v_{3}v_{2}v_{1}}\right)\geq 0$ **then**7:           [V,d]← S1D($\left\{v_{3},v_{1}\right\}$)8:        **else**9:           **if** $v_{1}^{T}n_{v_{3}v_{2}v_{1}}\geq 0$ **then**10:               $V\leftarrow \{v_{3},v_{2},v_{1}\}$11:               $d\leftarrow -n_{v_{3}v_{2}v_{1}}$12:           **else**13:               $V\leftarrow \{v_{3},v_{2},v_{1}\}$14:               $d\leftarrow n_{v_{3}v_{2}v_{1}}$15:           **end if**16:        **end if**17:    **end if**18:
**end function**



**Algorithm 5 **Sub-routine for |V|=4
1:**function**S3D($\left\{v_{4},v_{3},v_{2},v_{1}\right\}$)2:    $n_{v_{3}v_{2}v_{1}}=v_{12}\times v_{13}$3:    **if** $\left(v_{1}^{T}n_{v_{3}v_{2}v_{1}}\right)\left(v_{14}^{T}n_{v_{3}v_{2}v_{1}}\right)\geq 0$ **then**4:        [V,d]← S2D($\left\{v_{3},v_{2},v_{1}\right\}$)5:    **else**6:        $n_{v_{4}v_{3}v_{1}}=v_{13}\times v_{14}$7:        **if** $\left(v_{1}^{T}n_{v_{4}v_{3}v_{1}}\right)(v_{12}^{T}n_{v_{4}v_{3}v_{1})}\geq 0$ **then**8:           [V,d]← S2D($\left\{v_{4},v_{3},v_{1}\right\}$)9:        **else**10:           $n_{v_{4}v_{2}v_{1}}=v_{12}\times v_{14}$11:           **if** $\left(v_{1}^{T}n_{v_{4}v_{2}v_{1}}\right)\left(v_{13}^{T}n_{v_{4}v_{2}v_{1}}\right)\geq 0$ **then**12:               [V,d]← S2D($\left\{v_{4},v_{2},v_{1}\right\}$)13:           **else**14:               $V\leftarrow \{v_{4},v_{3},v_{2},v_{1}\}$15:           **end if**16:        **end if**17:    **end if**18:
**end function**



## 3. Bézier Curves

### 3.1. Continuity Conditions

As explained before, at each sampling time, a trajectory, expressed as a parametric Bézier curve, is generated for the time horizon $\left[t_{k},t_{k}+t_{h}\right]$, and the trajectory for the entire flight time [0,T] is formed by joining segments of these Bézier curves end-to-end. The smoothness of the resulting composite trajectory must be guaranteed by enforcing continuity at the joining points of two consecutive segments up to a certain derivative. In the following, in order to derive conditions that address parameter continuity between consecutive curves, we assume that the time horizon is equal to $\Delta t_{k}=t_{k+1}-t_{k}$, which is not necessarily the same for all sub-problems. In practice, however, the time horizon is greater than $\Delta t_{k}$, in which case a Bézier curve describing the segment over the time interval $\left[t_{k},t_{k+1}\right]$ can be obtained by subdividing $p_{k}\left(.\right)$ at $t_{k+1}$ with the de Casteljau’s algorithm. For simplicity we drop the subscript $i\in [N_{v}]$.

The Bézier curve describing the trajectory over the time interval $\left[t_{k},t_{k+1}\right]$ is defined as
(34)$$\begin{array}{c} p_{k}\left(\tau_{k}\right)=\sum_{l=0}^{n_{k}}{\bar{p}}_{l,k}B_{l,n_{k}}\left(\tau_{k}\right), \end{array}$$

Assuming that the global parameter *t* runs over the interval $\left[t_{k},t_{k+1}\right]$, the local parameter $\tau_{k}$ is related to *t* by
(35)$$0\leq \tau_{k}=\frac{t-t_{k}}{t_{k+1}-t_{k}}\leq 1$$

The parametric continuity condition, $C^{r}$, requires the r-th derivative and all lower derivatives of two consecutive segments to be equal at the joining point. In other words,
(36)$$\frac{d^{r}p_{k}\left(1\right)}{dt^{r}}=\frac{d^{r}p_{k+1}\left(0\right)}{dt^{r}}\;r\in \left\{0,\cdots ,r\right\}$$

Zero-order parametric continuity, $C^{0}$, requires the endpoints of two consecutive curves, $p_{k}\left(.\right)$ and $p_{k+1}\left(.\right)$, to intersect at one endpoint, that is,
(37)$$p_{k}\left(1\right)=p_{k+1}\left(0\right)$$

Since a Bézier curve is coincident with its control points at the two ends, i.e.,
(38)$$p_{k}\left(0\right)={\bar{p}}_{0,k}\;p_{k}\left(1\right)={\bar{p}}_{n_{k},k},$$
the position continuity condition (Equation 37) translates into
(39)$${\bar{p}}_{n_{k},k}={\bar{p}}_{0,k+1}$$

The first-order parametric continuity condition, $C^{1}$, for the *k*-th and k+1-th Bézier curves, can be obtained as
(40)$$\begin{array}{cc} \Delta t_{k+1}n_{k} & ({\bar{p}}_{n_{k},k}-{\bar{p}}_{n_{k}-1,k})=\\ \; & \Delta t_{k}n_{k+1}\left({\bar{p}}_{1,k+1}-{\bar{p}}_{0,k+1}\right) \end{array}$$

Finally, the *k*-th and k+1-th Bézier curves are $C^{2}$-continuous if
(41)$$\begin{array}{cc} \frac{\Delta t_{k+1}}{\Delta t_{k}} & \left({\bar{p}}_{n_{k}-1,k}-{\bar{p}}_{n_{k}-2,k}\right)\\ \; & +n_{k}\left(n_{k+1}-1\right){\bar{p}}_{n_{k}-1,k}+n_{k}{\bar{p}}_{n_{k},k}=\\ \frac{\Delta t_{k}}{\Delta t_{k+1}} & \left({\bar{p}}_{1,k+1}-{\bar{p}}_{2,k+1}\right)\\ \; & +n_{k+1}\left(n_{k}-1\right){\bar{p}}_{1,k+1}+n_{k+1}{\bar{p}}_{0,k+1} \end{array}$$

Higher-order parametric continuity conditions can be obtained likewise.

### 3.2. Evaluating Inequalities in Bézier Form

Parameterizing the trajectory with a Bézier curve converts the original infinite dimensional problem (Equation 3) into a semi-infinite optimization problem with a finite number of decision variables and an infinite number of constraints. The commonly used approach to obtaining a standard optimization problem is time gridding, which inspects satisfaction of constraints on a finite number of points only. Although this method is straightforward, it cannot guarantee that constraints are satisfied over the entire time interval. Using fine discretization can remedy this issue, but, it will increase the number of constraints as well as the computation time. Since all constraints involved in the trajectory generation problem addressed above can be expressed as Bézier curves, we can employ the method proposed in [33] to recast the semi-infinite optimization problem into one that is computationally tractable. As explained below this method exploits unique features of Bézier curves to efficiently evaluate constraints while avoiding problems associated with time gridding.

If h(τ) can be expressed as a Bézier curve, then any inequality constraint of the form h(τ)≤0,τ∈[0,1] can be replaced by a finite set of constraints on its control points. More specifically, if h(τ) is defined as
(42)$$\begin{array}{c} h\left(\tau \right)=\sum_{l=0}^{n_{h}}{\bar{h}}_{l}B_{l,n_{h}}\left(\tau \right), \end{array}$$
then from the *convex hull* property of Bézier curves we know that
(43)$$h\left(\tau \right)\in CH\left(\bar{H}\right)\;\tau \in \left[0,1\right]$$
where $CH\left(\bar{H}\right)=\left\{\alpha_{0}{\bar{h}}_{0}+\cdots +\alpha_{n_{h}}{\bar{h}}_{n_{h}}|\alpha_{0}+\cdots ,\alpha_{n_{h}}=1,\alpha_{l}\geq 0\right\}$ is the convex hull defined by the set of control points [34]. Thus, the inequality constraint h(τ)≤0 holds if
(44)$$\begin{array}{c} {\bar{h}}_{l}\leq 0\;for\;l=0,\cdots ,n_{h}. \end{array}$$

This finite set of inequality constraints can ensure that the original inequality constraint is satisfied over the entire interval [0,1]. However, Ineqs. (Equation 44) might be conservative due to the existing gap between the control points ${\bar{h}}_{l}$ and the actual curve h(τ). This problem can be alleviated by refining the control polygon and finding closer control points to the curve using recursive subdivision of h(τ) with the de Casteljau’s algorithm. The sequence of control polygons generated with repeated subdivision converges to the underlying Bézier curve [35]. Figure 6 shows a threefold subdivision of a cubic Bézier curve. Furthermore, the de Casteljau’s algortithm allows refining the control polygon locally. Using recursive subdivsion of h(τ) to reduce the conservatism in the finite set of constraints results in an increase in the number of constraints; hence, a trade-off has to be made between the computational effort and the conservatism. Nevertheless, the optimization variables remain the same [36].

## 4. Simulation Results

In this section, the efficacy of the proposed method for generating feasible and collision-free trajectories in (vehicle-) dense environments are assessed through different simulation examples. We compare the resulting trajectories to those generated with the well-studied BVC approach. We specifically test the capability of the two methods to generate trajectories that ensure all drones involved in a simulation example reach their final positions, and compare the flight time, obtained with each of them, to complete point-to-point transition missions. We also present the recorded computation time for executing the proposed algorithm in this paper to emphasize its suitability for real-time applications.

In the simulations presented below, we assume all drones have the same size, and their BVC (Equation 23) is defined with the safety radius $r_{D}=0.30m$. To specify the set (Equation 25), we approximate the drone body with an oblate spheroid with Λ=diag([0.30m,0.30m,0.11m]). In both methods, trajectories are parameterized with Bézier curves. Upper and lower bounds on the speed and acceleration are assumed to be $\pm 2.3\;\frac{m}{s}$ and $\pm 7.1\;\frac{m}{s^{2}}$ respectively. At each replanning step, the planner finds the closest point in the updated Voronoi cell to the goal position using the algorithm in Section 2.3. The computed point is then used to define the terminal cost term. The first term of the cost function in all subproblems is defined as $\int_{0}^{1}{∥p_{i,k}^{\left(4\right)}\left(\tau \right)∥}^{2}d\tau$. The time horizon and the replanning step are also considered to be the same for both methods. The obtained solution at the previous replanning step is used to set the initial guess for the current sub-problem. We use FORCES Pro [37] to generate solvers for the resulting sub-problems. The sub-problems, involving the set of control points ${\bar{p}}_{i,k}$ as decision variables, can be reformulated to match the supported classes of problem in FORCES Pro. Here, all computations are executed on a single desktop computer, with 2.60 GHz i7-4510U CPU and 6.00 GB RAM; however, in practice, the resulting independent sub-problems can be solved in parallel.

As mentioned before in the paper, in Voronoi-based methods, a vehicle only requires the position information from its neighboring vehicles to generate its trajectory. Therefore, they are more suitable for implementation when vehicles have limited communication capability, and have to rely solely on onboard sensing. In reality, the position sensor noise can impact the planner performance, yet this is more pronounced when estimating other information, such as velocity, from noise-corrupted measurements is needed. Therefore, Voronoi-based planners are more robust when there is no communication network. Nevertheless, in the following simulations, we assume that accurate position information is available with no delay at the replanning time.

In the first example, we consider five drones flying from their initial positions to given final positions. This example is similar to one in [24] where a random offset is added to break the symmetry in the drones’ initial and final configurations. Figure 7 (right) shows collision-free trajectories generated with the distributed scheme described above, with a replanning rate of 20 Hz. For this particular example, the resulting trajectories match those generated with BVC with a flight time of 11.6782 s. Figure 7 (left) shows collision-free trajectories obtained from the centralized solution, which delivers a total flight time of 9.4347 s, yet, while the central solution is obtained in 601 ms, the average computation time for solving the sub-problems in the decentralized scheme is only 49 ms.

In the next example, we consider 18 drones switching positions in a 3 m×5 m×2 m space, with a maximum speed and acceleration of $\pm 4.7\;\frac{m}{s}$ and $\pm 9.8\;\frac{m}{s^{2}}$, respectively. Figure 8a shows the initial and final configurations, and Figure 8b displays collision-free trajectories generated with the proposed distributed scheme in the paper implemented at 10 Hz. While both methods could find collision-free trajectories for guiding the team of drones from their initial positions to their goal positions, the flight time achieved with the proposed method is markedly shorter than the time obtained with BVC. We also performed a trial simulation with 34 drones in a similar configuration. Table 1 compares the success rate and the flight time to complete the transition using BVC and the proposed method.

In the third example, we consider 100 drones flying in an 8 m×8 m×3.5 m space. The initial and final positions for the drones are displayed with dot and square markers in Figure 9. We test both methods in 30 different trials. In each trial, final positions are randomly assigned to drones. A trial is considered successful if all drones could reach their final positions within the stipulated time. The proposed method with 23 successful trials and an average total flight of 1s outperforms the BVC with only 16 completed trials. It should be noted that using well-devised deadlock prevention strategies or loosening time constraints can improve the success rate of both methods. Figure 9 shows collision-free trajectories generated with the proposed method for one of the trials at different time steps. The average computation time for solving sub-problems in this example was around 115 milliseconds. In addition, compared to the geometric algorithm in [24], the closest point in a Voronoi cell to the goal position was obtained at least 10 times faster with the proposed algorithm in Section 2.3. The computation time for finding the closest point, and solving the optimization problem, depends on the number of neighboring drones (See Table 2 for recorded computation times in simulation examples with 18, 34, and 100 drones). In most applications, with typical Voronoi diagrams, the number of boundary planes, i.e., the number of Voronoi neighbors, is small. Thus, the proposed distributed algorithm is scalable to arbitrary numbers of vehicles.

## 5. Conclusions

In this paper, we introduce an efficient distributed algorithm for generating collision-free trajectories for multiple drones, taking into account their orientation. In order to avoid substantial communication between drones, we adopt Voronoi-based space partitioning and derive local constraints that guarantee collision avoidance with neighboring vehicles for an entire time horizon. We leverage Bézier curve properties to ensure that the set of collision avoidance constraints are satisfied at any time instant of the planning horizon. The same approach can be employed to obtain local collision avoidance constraints for the cases where the normal vector and offset of separating planes are time-varying parameters or decision variables of sub-problems. Yet, adopting Voronoi diagram with fixed planes for an entire planning horizon, though being conservative, results in simple, small sub-problems allowing for the trajectories to be replanned at a higher rate. We present different simulation results to highlight the scalability of the algorithm to large numbers of drones, and also its capability to generate less conservative trajectories with notably shorter flight times, compared to other Voronoi-based methods.

Our future work includes implementation and experimental validation of the algorithm for teams of drones. As we explain in the paper, at each time sample, upon receiving (or sensing) the new position information, a vehicle must find the closest point in its Voronoi cell to the goal position, and solve an optimization problem that uses the current state of the vehicle as the initial condition, to generate its trajectory for a certain time horizon. Although the time to compute the closest point is mainly negligible, the computation time to find the optimal solution can lead to a (significant) delay between updating the position information and executing the trajectory. Therefore, in practice, the computational delay must be explicitly considered to avoid performance degradation (or even failure) of the planner.

## Figures and Tables

**Figure 1 sensors-22-01855-f001:**
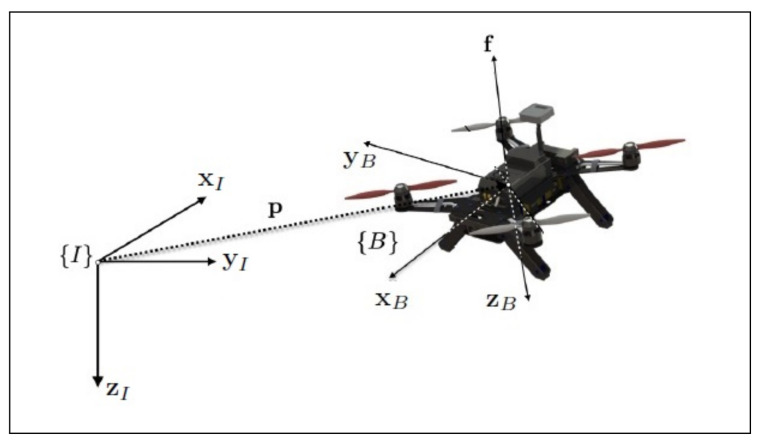
The quadrotor reference frames.

**Figure 2 sensors-22-01855-f002:**
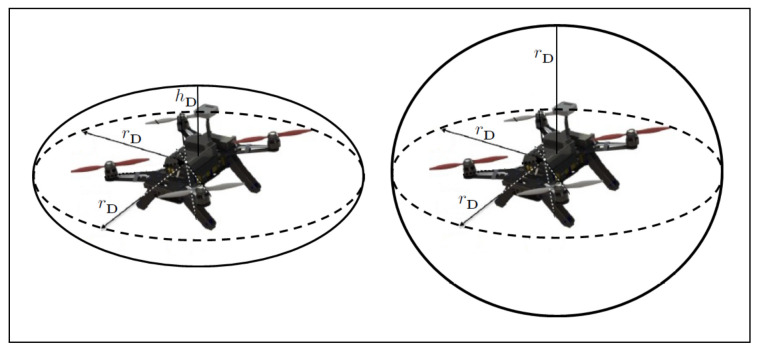
The quadrotor body can be represented as a sphere with radius $r_{D}$ (**right**), or an ellipsoid aligned with the axes of the body frame (**left**). Approximating the drone body with an ellipsoid allows considering the quadrotor’s rotational motion.

**Figure 3 sensors-22-01855-f003:**
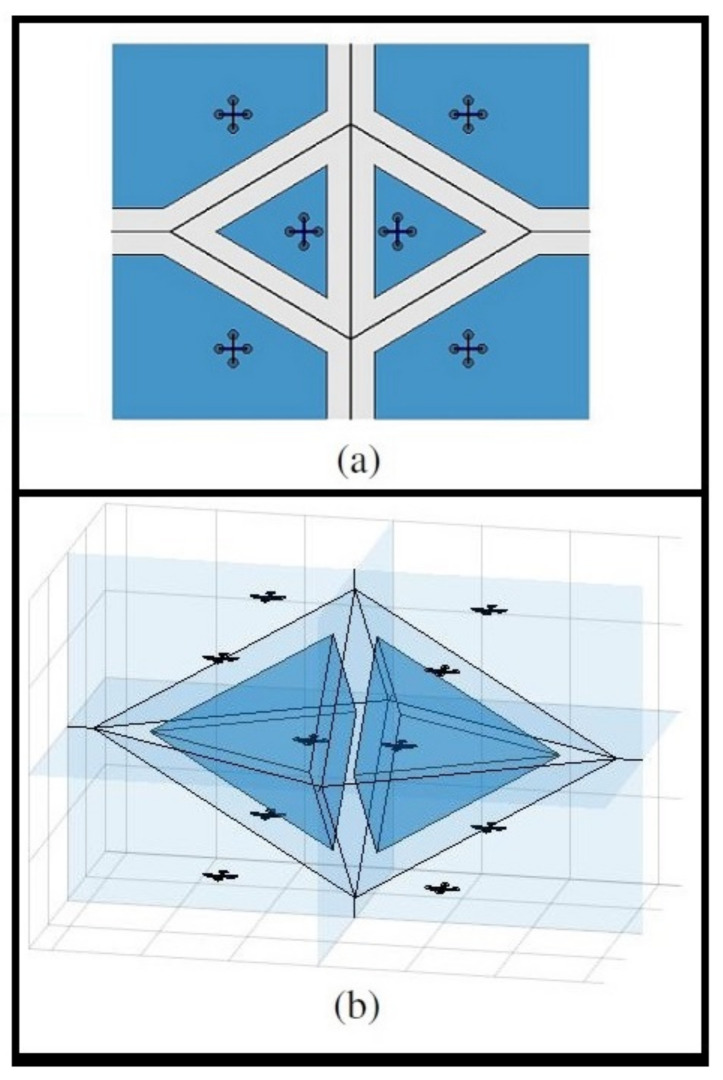
(**a**) The Voronoi diagram for six drones in 2D space. The Voronoi boundary edges are shown with solid black lines, and the buffered Voronoi cells are shaded in dark blue. (**b**) The Voronoi diagram for 10 drones in a collision-free configuration in 3D space. The Voronoi boundary ∂V is shaded in light blue, and the buffered Voronoi cells $\bar{V}$ for two neighboring drones in the center are shown in dark blue.

**Figure 4 sensors-22-01855-f004:**
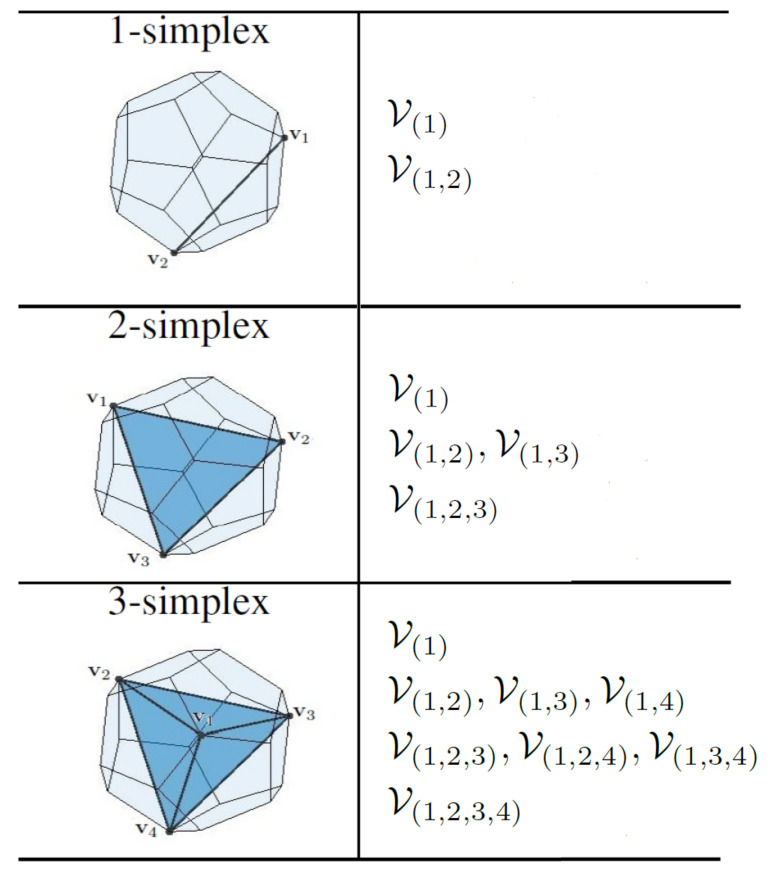
An m-simplex is linked to $2^{m+1}-1$ Voronoi regions associated with its vertices, edges, faces, and volume. The list of $2^{m}$ Voronoi regions that can possibly contain the origin is given in this table. It should be noted that $v_{1}$ is the latest vertex added to *V*.

**Figure 5 sensors-22-01855-f005:**
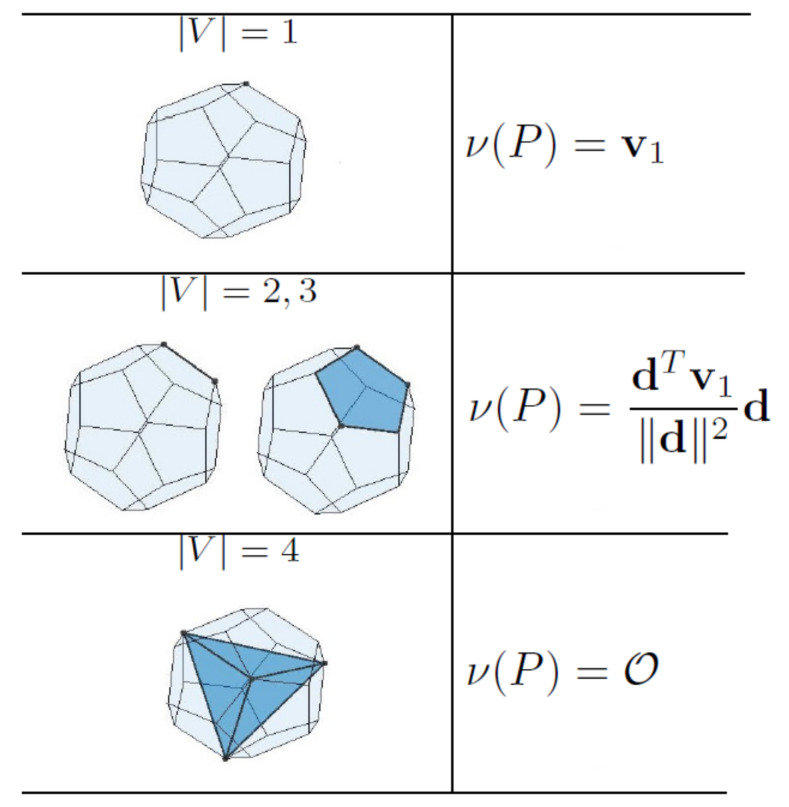
Examples of the closest feature of a polyhedron to a query point are shown above. Once the closest feature is obtained from Algorithm 1, the closest point, i.e., ν(P), can be determined, as shown above.

**Figure 6 sensors-22-01855-f006:**
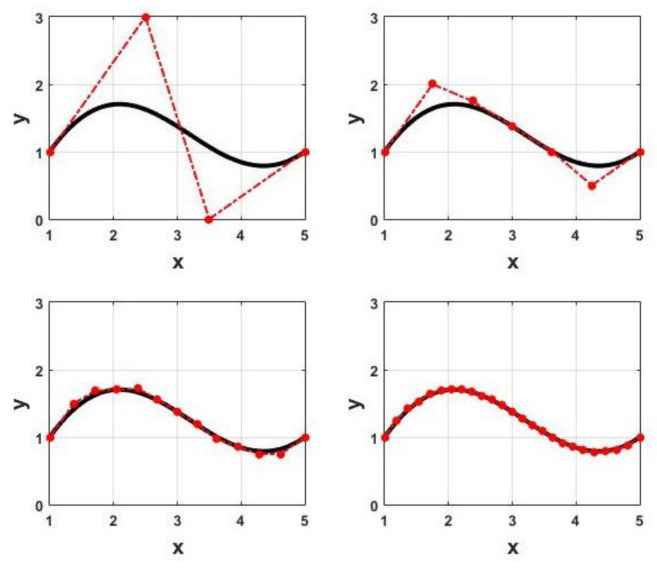
A cubic Bézier curve (**top left**) is subdivided into two Bézier curves of the same degree (**top right**) using the de Casteljau’s algorithm. The control polygon generated by recursive subdivision converges to the original Bézier curve ref. [28]. Copyright 2021 IEEE. Successive refinement of the original control polygon after 2 (**bottom left**) and 3 (**bottom right**) subdivisions.

**Figure 7 sensors-22-01855-f007:**
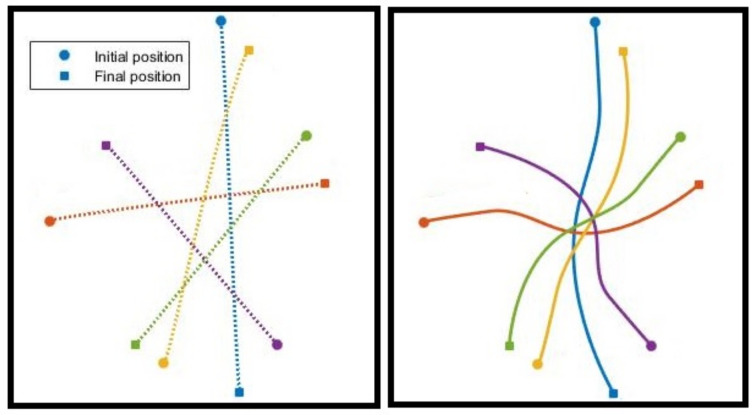
Comparing collision-free trajectories generated with the centralized solution (**left**) and the proposed decentralized approach (**right**) for five drones flying from their initial positions to given final positions. While the central solution yields a shorter flight time, its computation time is significantly longer than average time required to solve the sub-problems in the distributed method.

**Figure 8 sensors-22-01855-f008:**
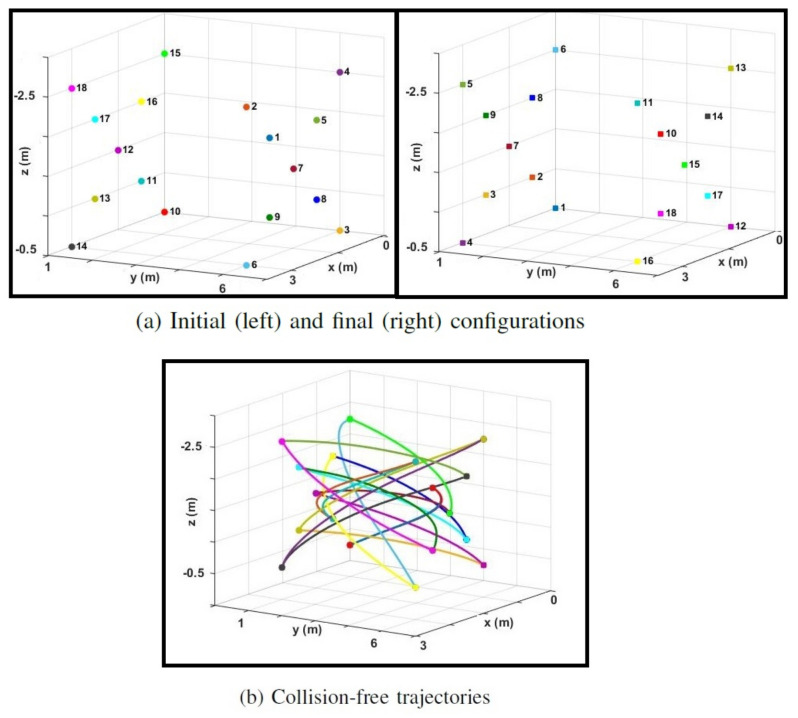
(**a**) Initial (left) and final (right) position configurations for 18 drones. Each drone is assigned a unique color and a number next to it. (**b**) Collision−free trajectories for 18 drones switching their positions in a 3 m×5 m×2 m space. The total flight time for the drones to reach their final positions is 5.1 s using the proposed method, which is shorter than the 6.3 s flight time obtained with the BVC.

**Figure 9 sensors-22-01855-f009:**
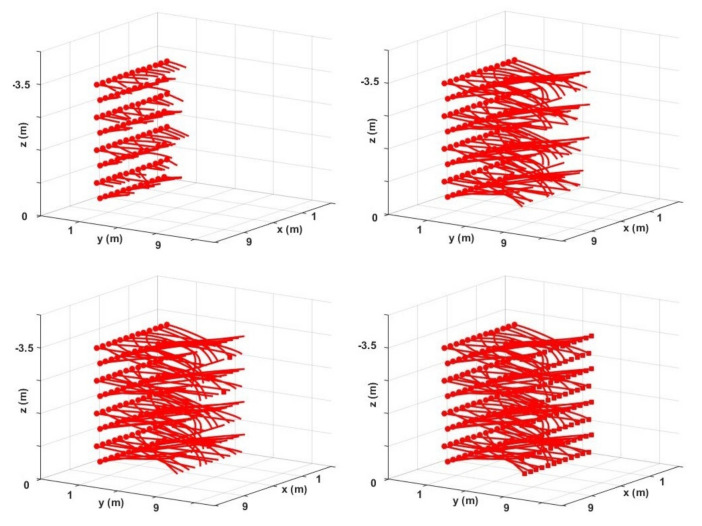
Collision−free trajectories for 100 drones flying from their initial positions (dots) to randomly specified final positions (squares) at different replanning steps.

**Table 1 sensors-22-01855-t001:** Comparing the number of successful trials and the average flight time achieved with the BVC and the proposed method in the paper.

Number of Drones	BVC		Proposed Method	
	Flight Time	Completed Trials	Flight Time	Completed Trials
18	6.812 s	5/5	5.327 s	5/5
34	8.105 s	7/10	6.625 s	8/10
100	14.573 s	16/30	11.462 s	23/30

**Table 2 sensors-22-01855-t002:** Recorded computation times for finding the closest point in a Voronoi cell to the goal position and solving the optimization problem in simulation examples with 18, 34, and 100 drones.

Number of Drones	Computation Time (ms)	
	Finding the Closest Point	Solving the Sub-Problem
18	<0.1	77.562
34	<0.1	98.330
100	0.171	121.633

## Data Availability

Not applicable.

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
