# Peer review of "A Distributed Algorithm for Real-Time Multi-Drone Collision-Free Trajectory Replanning"

_sensors, 2022, doi:10.3390/s22051855_

Round 1
Reviewer 1 Report
Dear Author,
thanks a lot for this well prepared paper. From my point of view, it is very well done including the scientific soundness, description of the challenges and the presentation of the proposed solution. Therefore I only have some minor points I would like to address here:
- line 183: should read "Problem formulation"
- equation (2): I did not find an explanation for L and its meaning in the context of this paper. Maybe I have missed it. The same hold for eq. (4).
- line 198: Why do you align the thrust force with the body's z axis? If a multicopter UAV moves, this usually does not hold.
- The same point should be addressed with respect to eq. (17): How do you take into account the orientation of the ellipsoid during the phases of acceleration?
- Fir. 3: The locations of the drones in 3D space are not/not easily identifiable.
- The phrase of quintic Bézier curve is used only in the caption of fig. 6. It should be mentioned in the text.
- lines 475-487: This paragraph does not make sense in a chapter describing the simulation results and should either go into the conclusions or the introduction.
- The text in lines 493-496 contradicts the caption of fig 7 (right hand and left hand sides swapped?)
Best regards!
Reviewer 2 Report
This is an interesting and well-written paper. The procedure is described in detail and comprehensible. In the end, I only have a few suggestions for changes
Lines 10 – 12: Specify the quantity of UAVs used for the simulation. What is the benchmark algorithm? Can the achieved improvement be quantified?
Lines 17 – 19: A differentiation between path planning and trajectory planning would be useful.
Lines 32 – 33: “…a multitude of distributed schemes have been proposed …” can you give some examples
Lines 37 – 39: Some examples here as well
Line 43: What does “active coupling” mean in this context?
Line 58: What does ADMM stand for?
Lines 133 – 134: Is only the current position of the neighboring UAVs used for planning or also their variation in position over time, e.g. estimated by a Kalman filter?
Line 140: Are other (static) obstacles considered in the operating space?
Line 162: For the sake of completeness, also specify the intervals of i and j, later also k
Line 191: How does the position of the other UAVs change during the planning period t_h?
Line 236: Provide references as examples
Line 298: Does this mean the UAV can move freely in its Voronoi cell? How is the change of the position of the other UAVs over the planning period t_h taken into account?
Line 395: Why are Bezier Curves used? What could be the alternatives? e.g. Cubic / Quintic Splines ...
Line 419: Which degree of continuity is aimed at for the Bezier Curves in this work? C2 as well?
Line 451: How long is the planning horizon or the number of planning time steps in your simulation?
Line 490: What would happen in the case of symmetry?
Lines 488 – 496: Is trajectory planning with BVC performed as often as the algorithm presented here? Are the constraints of the trajectory generation the same / similar? Can you explain in more detail the reason for the time difference.
Fig. 8a: The change of the UAV positions is difficult to see. Better: Use arrows to indicate the change in position. Add axis labels
Fig. 9: The representation of the change of the UAV positions is difficult to see. A revision would be useful here.
Author Response
Please see theattachment.
